# Conversion of a conventional superconductor into a topological superconductor by topological proximity effect

C.X. Trang[1], N. Shimamura[1], K. Nakayama[1,2], S. Souma[3,4], K. Sugawara[1,3,4], I. Watanabe[1], K. Yamauchi[5], T. Oguchi[5], K. Segawa[6], T. Takahashi[1,3,4], Yoichi Ando[7] & T. Sato[1,3,4]*

Realization of topological superconductors (TSCs) hosting Majorana fermions is a central challenge in condensed-matter physics. One approach is to use the superconducting proximity effect (SPE) in heterostructures, where a topological insulator contacted with a superconductor hosts an effective $p$-wave pairing by the penetration of Cooper pairs across the interface. However, this approach suffers a difficulty in accessing the topological interface buried deep beneath the surface. Here, we propose an alternative approach to realize topological superconductivity without SPE. In a Pb(111) thin film grown on TlBiSe$_2$, we discover that the Dirac-cone state of substrate TlBiSe$_2$ migrates to the top surface of Pb film and obtains an energy gap below the superconducting transition temperature of Pb. This suggests that a Bardeen-Cooper-Schrieffer superconductor is converted into a TSC by the topological proximity effect. Our discovery opens a route to manipulate topological superconducting properties of materials.

[1] Department of Physics, Tohoku University, Sendai 980-8578, Japan. [2] Precursory Research for Embryonic Science and Technology (PRESTO), Japan Science and Technology Agency (JST), Tokyo 102-0076, Japan. [3] Center for Spintronics Research Network, Tohoku University, Sendai 980-8577, Japan. [4] WPI Research Center, Advanced Institute for Materials Research, Tohoku University, Sendai 980-8577, Japan. [5] Institute of Scientific and Industrial Research, Osaka University, Ibaraki, Osaka 567-0047, Japan. [6] Department of Physics, Kyoto Sangyo University, Kyoto 603-8555, Japan. [7] Institute of Physics II, University of Cologne, Köln 50937, Germany. *email: t-sato@arpes.phys.tohoku.ac.jp

Topological superconductors (TSCs) are a peculiar class of superconductors where the nontrivial topology of bulk leads to the emergence of Majorana bound states (MBSs) within the bulk superconducting gap[1–5]. Since MBSs are potentially applicable to the fault-tolerant quantum computation, searching for a new type of TSCs is one of the central challenges in quantum science. A straightforward way to realize TSCs would be to synthesize an odd-parity $p$-wave superconductor; however, intrinsic $p$-wave pairing is rare in nature, as highlighted by a limited number of $p$-wave superconductor candidates hitherto reported (e.g., refs. [6,7]). A different approach to realize TSCs is to utilize the superconducting proximity effect (SPE) in a heterostructure consisting of a conventional superconductor and a spin-orbit coupled material such as a topological insulator (TI), as initiated by the theoretical prediction of effectively $p$-wave superconductivity induced in helical Dirac fermions and Rashba states[8,9]. This approach has been widely applied to various superconducting hybrids[10–17], whereas the existence of MBSs is still under intensive debates. A part of the difficulty in establishing the SPE-derived topological superconductivity may lie in the SPE process itself, since the searched MBSs are expected to be localized in the vortex core at the interface within the heterostructure, and hence are hard to be accessed by surface-sensitive spectroscopies such as scanning tunneling microscopy (STM). Therefore, it would be desirable to invent an alternative way to realize TSCs without using bulk $p$-wave superconductor or the SPE.

In this work, we present the possibility to realize TSCs by using the topological proximity effect (TPE)[18]; such a novel approach was discovered through our angle-resolved photoemission (ARPES) study of a heterostructure consisting of an epitaxial Pb thin film grown on a three-dimensional (3D) TI, TlBiSe$_2$.

## Results

**Fabrication and characterization of Pb film on TlBiSe$_2$.** The studies of SPE for generating TSCs have often employed a heterostructure consisting of a TI thin film as a top layer and a Bardeen-Cooper-Schrieffer (BCS) superconductor as a substrate[14–17]. On the other hand, in our TPE approach, the stacking sequence is reversed, and a superconducting Pb thin film was grown on TlBiSe$_2$ (Fig. 1a). We have deliberately chosen this combination, because (i) Pb films are known to maintain the superconductivity down to a few monolayers (MLs)[19], and (ii) TlBiSe$_2$ serves as a good substrate for epitaxial films[18]. Using the low-energy-electron-diffraction (LEED) (inset to Fig. 1d, f, h) and the ARPES results, we have estimated the in-plane lattice-constant $a$ to be 3.5 and 4.2 Å for Pb (~20 ML) and TlBiSe$_2$, respectively. While the $a$ value of Pb film is close to that of bulk[20], there is a sizable lattice mismatch of 19.5% between the Pb film and TlBiSe$_2$.

First, we discuss the overall electronic structure. As shown in Fig. 1c, d, the electronic band structure of pristine TlBiSe$_2$ is characterized by a Dirac-cone surface state (SS) around the $\bar{\Gamma}$ point that traverses the bulk valence and conduction bands[21–23], forming a small Fermi surface (FS) centered at $\bar{\Gamma}$. Upon evaporation of Pb on TlBiSe$_2$, the electronic structure drastically changes as seen in Fig. 1e, f; the holelike valence band of TlBiSe$_2$ disappears, while several M-shaped bands emerge. The outermost holelike band crosses the Fermi level ($E_F$) and forms a large triangular FS (Fig. 1e). The M-shaped bands are ascribed to the quantum well states (QWSs) due to the quantum confinement of electrons in the Pb thin film. This is supported by the experimental fact that similar M-shaped bands are also observed in a Pb(111) thin film grown on Si(111) (Fig. 1h).

The QWSs in Pb thin films with various thickness on Si(111) have been well studied by spectroscopies and calculations[19,24–28]. Since the in-plane lattice constant of Pb/TlBiSe$_2$ is close to that of Pb/Si(111), we expect a similar electronic structure between the two. By referring to the previous studies and our band-structure calculations, we estimated the film thickness to be 17 ML for the case in Fig. 1e, f; see Supplementary Fig. 1 and Supplementary Note 1. We observed no obvious admixture from other MLs (e.g., 16 and 18 MLs) that would create additional QWSs[24,26], suggesting an atomically flat nature of our Pb film. The LEED pattern of 17ML-Pb/TlBiSe$_2$ as sharp as that of pristine TlBiSe$_2$ (inset to Fig. 1f, d, respectively) also suggests the high crystallinity of Pb film. A careful look at Fig. 1f reveals an additional intensity spot near $E_F$ above the topmost M-shaped band. This band is not attributed to the QWSs, and is responsible for our important finding, as described below.

**Topological proximity effect.** Next we clarify how the band structure of TlBiSe$_2$ is influenced by interfacing with a Pb film. One may expect that there is no chance to observe the band structure associated with TlBiSe$_2$ because the Pb film (17 ML ~ 5 nm) is much thicker than the photoelectron escape depth (~0.5–1 nm). Figure 2a shows the ARPES-derived band structure near $E_F$ obtained with a higher resolution for 17ML-Pb/TlBiSe$_2$, where we clearly resolve an X-shaped band above the topmost QWSs. This band resembles the Dirac-cone SS in pristine TlBiSe$_2$ (Fig. 2c), and is totally absent in 17ML-Pb/Si(111) (Fig. 2b), thereby ruling out the possibility of its Pb origin. The appearance of a Dirac-cone-like band in 17ML-Pb/TlBiSe$_2$ is surprising, because the Pb film thickness is about ten times larger than the photoelectron escape depth. This in return definitely rules out the possibility that the observed Dirac-cone-like band is the Dirac-cone state embedded at the Pb/TlBiSe$_2$ interface. Furthermore, this band is not likely to originate from the accidentally exposed SS of TlBiSe$_2$ through holes in Pb, since the observed bands do not involve a replica of pristine TlBiSe$_2$ bands and no trace of the Tl core-level peaks was found in Pb/TlBiSe$_2$; see Supplementary Fig. 2 and Supplementary Note 2. In fact, the bulk valence band lying below 0.4 eV observed in pristine TlBiSe$_2$ (Fig. 2c) totally disappears in Pb/TlBiSe$_2$, and in addition, the Dirac point of Pb/TlBiSe$_2$ is shifted upward with respect to that of pristine TlBiSe$_2$, as clearly seen in Fig. 2d–f. These observations led us to conclude that the Dirac-cone band has migrated from TlBiSe$_2$ to the surface of Pb film via the TPE when interfacing Pb with TlBiSe$_2$[18]. Such a migration can be intuitively understood in terms of the adiabatic bulk-band-gap reversal[29,30] in the real space where the band gap (inverted gap) in TlBiSe$_2$ closes throughout the gapless metallic overlayer and starts to open again at the Pb-vacuum interface. It is noted that the upper branch of the Dirac-cone-like band would be connected to the quantized conduction band of the Pb film above $E_F$ because it only crosses $E_F$ once between $\bar{\Gamma}$ and M.

The band picture based on the TPE well explains the observed spectral feature in Pb/TlBiSe$_2$. As shown in Fig. 2g–i, the spin-degenerate topmost QWS of Pb (Fig. 2g) and the spin-polarized Dirac-cone SS of TlBiSe$_2$ (Fig. 2h) start to interact each other when interfacing Pb and TlBiSe$_2$. Due to the spin-selective band hybridization[18], the Dirac-cone band is pushed upward, while the QWS is pulled down (Fig. 2i). This is exactly what we see in Fig. 2a. Our systematic thickness-dependent ARPES measurements revealed a detailed hybridization behavior between the Dirac-cone band and the QWSs, supporting this scenario; see Supplementary Fig. 3 and Supplementary Note 3. Noticeably, the migration of Dirac-cone state is observed at least up to 22 ML thick (~6.5 nm thick) Pb film. Such a long travel of the Dirac cone

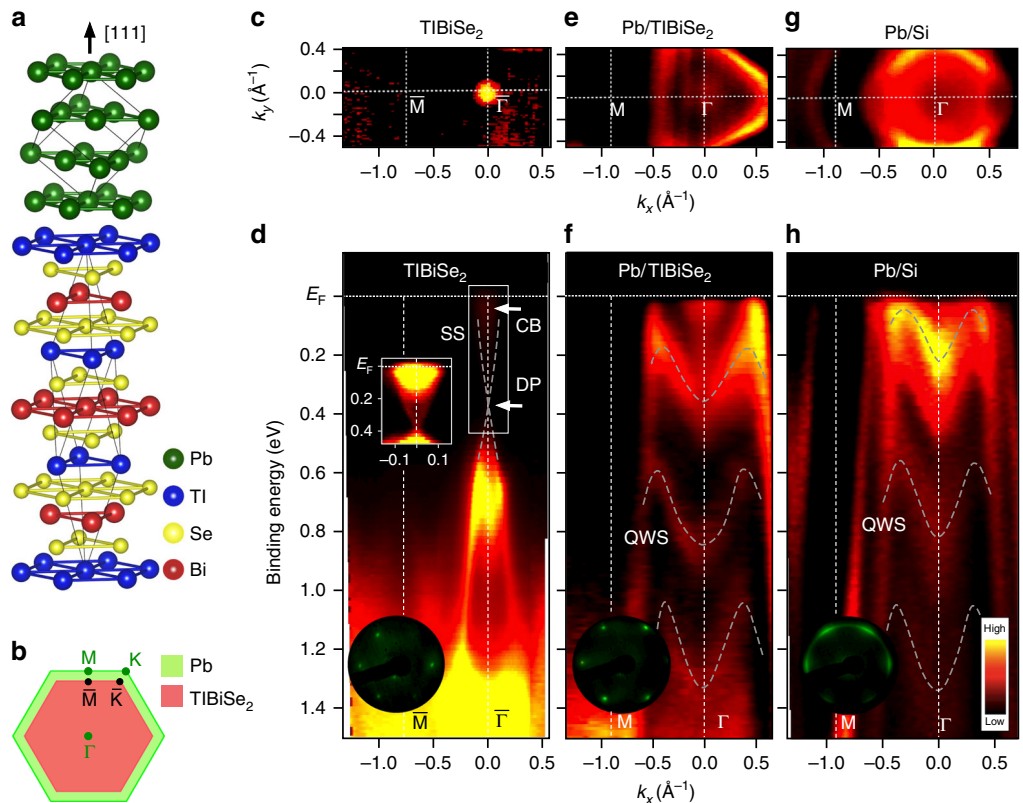

**Fig. 1 Crystal and electronic structures of Pb(111) thin film on TlBiSe₂. a** Heterostructure consisting of Pb and TlBiSe₂. **b** Comparison of Brillouin zone between TlBiSe₂ (red) and Pb (green). **c** Plot of ARPES intensity at $E_F$ as a function of in-plane wave vector (namely, Fermi surface) around the $\overline{\Gamma M}$ line for pristine TlBiSe₂, measured with the He-Iα line ($h\nu$ = 21.218 eV). **d** ARPES-derived band structure along the $\overline{\Gamma M}$ cut for pristine TlBiSe₂. Inset shows the ARPES intensity with enhanced color contrast in the area enclosed by white rectangle. **e**, **f** Same as **c** and **d** but for 17ML-Pb/TlBiSe₂. **g**, **h** Same as **e** and **f** but for 17ML-Pb/Si(111). SS, CB, DP, and QWS denote the surface state, conduction band, Dirac point, and quantum well state, respectively. Arrows in **d** indicate the location of CB and DP. Dashed gray curves are a guide for the eyes to trace the SS and QWSs. Insets to **d**, **f**, and **h** are the LEED patterns of the respective films.

in the real space is unexpected, and hard to be reproduced by the band calculations due to large incommensurate lattice mismatch between Pb and TlBiSe₂. In fact, we have tried to calculate the band dispersion of Pb/TlBiSe₂ slab by expanding the in-plane lattice constant of Pb film to hypothetically form a commensurate system, but it caused a sizable change in the whole band structure of Pb film, resulting in the band structure totally different from the experiment. Alternatively, a calculation that uses a larger in-plane unit cell might be useful to achieve an approximate lattice match between Pb and TlBiSe₂. It is noted here that the coherency of electronic states may play an important role for the observation of a coupling with the substrate (i.e., the TPE in this study) as in the case of other quantum composite systems involving metallic overlayer[31]. We estimate the electronic coherence length in Pb film to be larger than 22 ML (~6.5 nm) because the topological SS is observed even in the 22 ML film; see Supplementary Fig. 3 and Supplementary Note 3.

**Superconducting gap**. The next important issue is whether the Pb/TlBiSe₂ heterostructure hosts superconductivity. To elucidate it, we first fabricated a thicker (22 ML) Pb film on TlBiSe₂ and carried out ultrahigh-resolution ARPES measurements at low temperatures. Figure 3b shows the energy distribution curve (EDC) at the $\mathbf{k}_F$ point of the Pb-derived triangular FS (point A in Fig. 3a) measured at $T = 4$ and 10 K across the superconducting transition temperature $T_c$ of bulk Pb (7.2 K). At $T = 4$ K, one can clearly recognize a leading-edge shift toward higher $E_B$ together with a pile up in the spectral weight, a typical signature of the

superconducting-gap opening. This coherence peak vanishes at $T = 10$ K due to the gap closure, as better visualized in the symmetrized EDC (Fig. 3c). We have estimated the superconducting-gap size at $T = 4$ K to be 1.3 meV from the numerical fittings. This value is close to that of bulk Pb (~1.2 meV)[32], suggesting that the $T_c$ is comparable to that of bulk Pb.

Since the superconductivity shows up on the Pb film, we now address an essential question whether the migrating Dirac-cone band hosts superconductivity. We show in Fig. 3d–i the EDCs and corresponding symmetrized EDCs at $T = 4$ and 10 K for the 17 ML sample measured at three representative $\mathbf{k}_F$ points (points A–C in Fig. 3a). At point A on the Pb-derived FS, we observe the superconducting-gap opening (Fig. 3d), similarly to the 22 ML film. At point B (C), where the migrating Dirac-cone band crosses $E_F$ along the $\overline{\Gamma K}$ ($\overline{\Gamma M}$) line, we still observe a gap as seen in Fig. 3f (Fig. 3h). This indicates that an isotropic superconducting gap opens on the migrating Dirac-cone FS. We observed that this gap persists at least down to 12 ML, confirming that the superconducting gap is not an artifact that accidentally appears at some specific film thickness; see Supplementary Fig. 4 and Supplementary Note 4. We have also confirmed that the gap opening is not an inherent nature of the original topological SS in pristine TlBiSe₂, by observing no leading-edge shift or spectral-weight suppression at $E_F$ at 4 K in pristine TlBiSe₂ (Fig. 3j, k).

**Discussion**

The present results show that the superconducting gap opens on the entire FS originating from both the Pb-derived QWSs and the

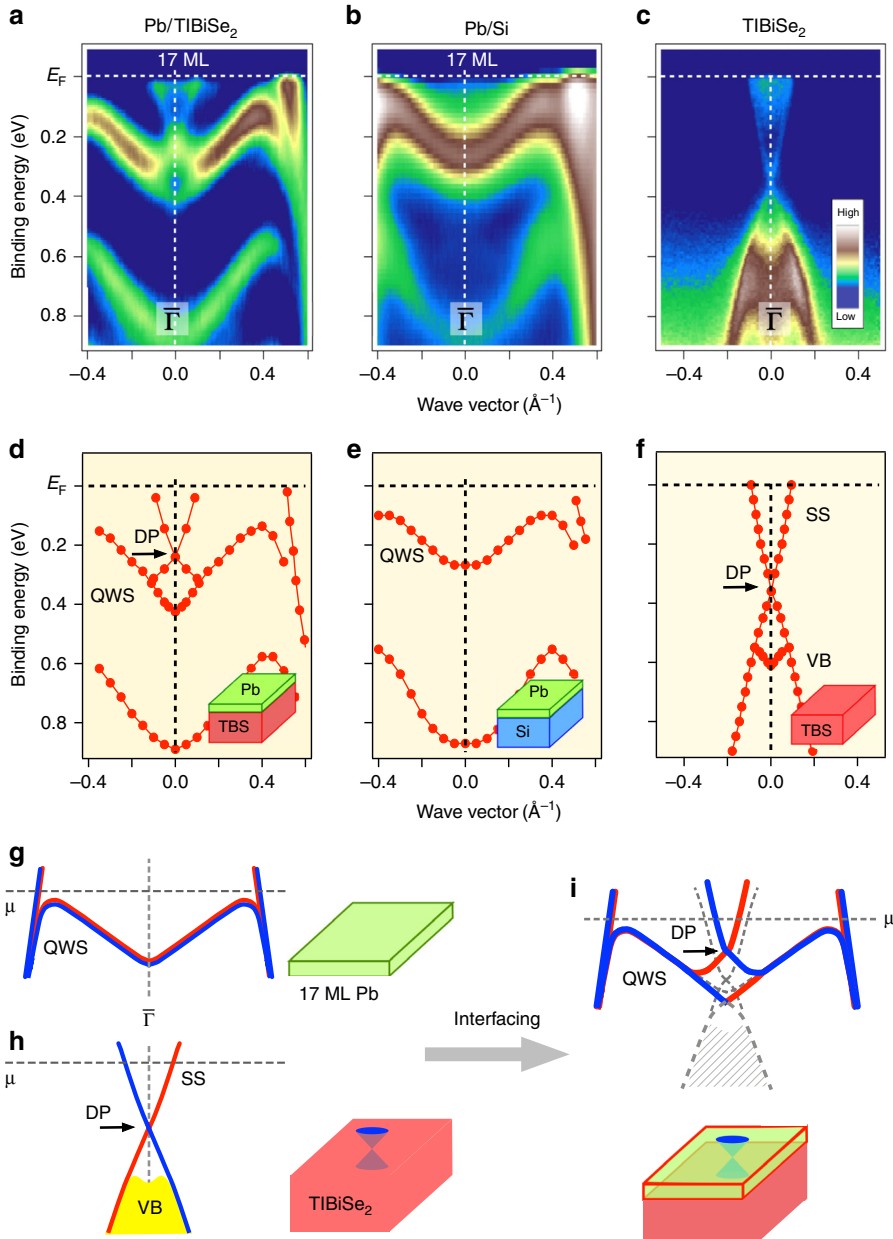

**Fig. 2 Migration of topological surface states to the surface of Pb film. a–c** ARPES-derived band structure near $E_F$ around the $\overline{\Gamma}$ point for 17ML-Pb/TlBiSe$_2$, 17ML-Pb/Si(111), and pristine TlBiSe$_2$, respectively, with the He-I$\alpha$ line. The data were obtained along the $\overline{\Gamma K}$ cut parallel to the analyzer slit, so that the energy and momentum resolutions are better than the data in Fig. 1d, f, h measured along the $\overline{\Gamma M}$ cut perpendicular to the analyzer slit. **d–f** Experimental band dispersions extracted from the peak positions of MDCs/EDCs for **a–c**. **g–i** Schematics of the hybridization between topological Dirac-cone state and QWSs, showing the migration of topological Dirac-cone state upon interfacing Pb with TlBiSe$_2$. Dashed curves in **i** indicate the band dispersion without hybridization. Note that the bulk VB of TlBiSe$_2$ indicated by gray shade becomes invisible on the Pb/TlBiSe$_2$ interface, because only the topological SS migrates to the top surface.

migrating Dirac-cone band (Fig. 3l). The emergence of an isotropic superconducting gap on the Dirac-cone FS suggests that the 2D topological superconductivity is likely to be realized, since this heterostructure satisfies the theoretically proposed condition for the effectively $p$-wave superconducting helical-fermion state[8]. In this regard, one may think that such realization is a natural consequence of making heterojunction between superconductor and TI. However, the present study proposes an essentially new strategy to realize the 2D topological superconductivity. In the ordinary approach based on the SPE (Fig. 3m), the topological Dirac-cone state hosts the effective $p$-wave pairing at the interface due to the penetration of Cooper pairs from the superconductor

to the TI. On the other hand, the present approach does not rely on this phenomenon at all, because the topological Dirac-cone state appears on the top surface of a superconductor (Fig. 3n) via the TPE.

One can view this effect as a conversion of a conventional superconductor (Pb film without topological SS) to a TSC (Pb film with topological SS) by interfacing. The present approach to realize 2D TSCs has an advantage in the sense that the pairing in the helical-fermion state (and the MBS as well) is directly accessed by surface spectroscopies such as STM and ARPES. The superconducting helical fermions would be otherwise embedded deep at the interface and are hard to be

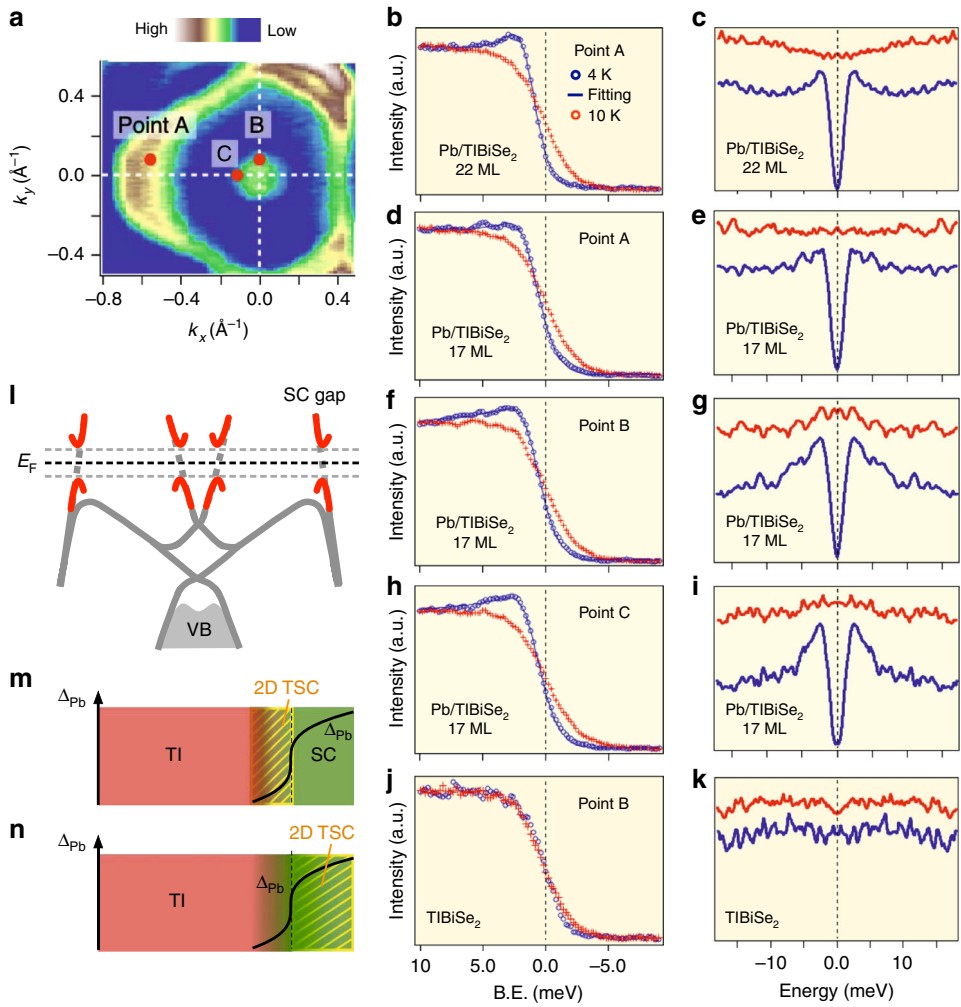

**Fig. 3 Possible topological superconductivity in Pb film. a** ARPES-intensity mapping at $E_F$ for 22ML-Pb/TlBiSe$_2$. **b, c** Ultrahigh-resolution EDCs and corresponding symmetrized EDCs, respectively, at $T = 4$ and 10 K, measured at point A in **a** ($\mathbf{k}_F$ point of the QWS) for 22ML-Pb/TlBiSe$_2$. **d, e** Same as **b** and **c**, respectively, but for 17ML-Pb/TlBiSe$_2$. **f, g** Same as **d** and **e**, respectively, but measured at point B ($\mathbf{k}_F$ point of the topological SS). **h, i** Same as **d** and **e**, respectively, but measured at point C. **j, k** Same as **f** and **g**, respectively, but for pristine TlBiSe$_2$. Blue solid curve in the EDC at $T = 4$ K in **b**, **d**, **f**, and **h** is the result of numerical fittings using the Dynes function multiplied by the Fermi–Dirac distribution function, convoluted with a resolution function. **l** Illustration of the superconducting-gap opening on the QWS- and TSS-derived bands. **m** Conventional view on the realization of 2D TSC via SPE. **n** Proposed method to realize 2D TSC by converting a conventional superconductor into a TSC through the TPE.

accessed if the TPE does not occur. Moreover, the observed gap magnitude on the topological SS is comparable to that of the original Pb, unlike the SPE-induced gap that is usually smaller. This result tells us that the so-far overlooked TPE had better be seriously taken into account in many superconductor-TI hybrids. Also, the present study points to the possibility of realizing even wider varieties of 2D TSCs by using the TPE. It is noted that the topological states in Pb/TlBiSe$_2$ are electrically shorted out by the metallic QWSs, unlike the case of some TI films on top of superconductors. This needs to be considered in the application because single conducting channel from the Dirac-cone states would be more preferable. In this regard, the present approach utilizing the TPE and the existing approach using the SPE would be complementary to each other.

## Methods

**Sample preparation**. High-quality single crystals of TlBiSe$_2$ were grown by a modified Bridgman method[21]. To prepare a Pb film, we first cleaved a TlBiSe$_2$ crystal under ultrahigh vacuum with scotch tape to obtain a shiny mirror-like surface, and then deposited Pb atoms (purity; 5 N) on the TlBiSe$_2$ substrate using the molecular-beam epitaxy technique while keeping the substrate temperature at $T = 85$ K. A Pb(111) film on Si(111), used as a reference, was fabricated by keeping

the same substrate temperature. The film thickness was controlled by the deposition time at a constant deposition rate. The actual thickness was estimated by a comparison of ARPES-derived band dispersions with the band-structure calculations for free-standing multilayer Pb.

**ARPES measurements**. ARPES measurements were performed with the MBS-A1 electron analyzer equipped with a high-intensity He discharge lamp. After the growth of Pb thin film by evaporation, it was immediately transferred to the sample cryostat kept at $T = 30$ K in the ARPES chamber, to avoid the clusterization of Pb that is accelerated at room temperature (note that such clusterization hinders the detailed investigation of the surface morphology by atomic-force microscopy). We used the He-I$\alpha$ resonance line ($h\nu = 21.218$ eV) to excite photoelectrons. The energy resolution of ARPES measurements was set to be 2–40 meV. The sample temperature was kept at $T = 30$ K during the ARPES-intensity-mapping measurements, while $T = 4$ and 10 K for the superconducting-gap measurements. The Fermi level ($E_F$) of the samples was referenced to that of a gold film evaporated onto the sample holder.

**Band calculations**. First-principles band-structure calculations were carried out by a projector augmented wave method implemented in Vienna Ab initio Simulation Package code[33] with generalized gradient approximation potential[34]. After the crystal structure was fully optimized, the spin-orbit coupling was included self-consistently.

## Data availability
The data sets generated/analyzed during the current study are available from the corresponding author on reasonable request.

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

## Acknowledgements
We thank T.N., K.H., A.T. and T.R. for their assistance in the ARPES experiments. This work was supported by Grant-in-Aid for Scientific Research on Innovative Areas "Topological Materials Science" (JSPS KAKENHI Grant numbers JP15H05853, JP18H04227, and JP15K21717), JST-CREST (no. JPMJCR18T1), JST-PRESTO (no. JPMJPR18L7), and Grant-in-Aid for Scientific Research (JSPS KAKENHI Grant numbers JP17H01139, JP15H02105, JP26287071, and JP25287079). The work in Cologne was funded by the Deutsche Forschungsgemeinschaft (German Research Foundation) - Project number 277146847 - CRC 1238 (Subproject A04).

## Author contributions
The work was planned and proceeded by discussion among C.X.T., K.N., S.S., K.S., T.S., T.T., and Y.A. K.S. carried out the growth of bulk single crystals and their characterization. C.X.T and I.W. fabricated ultrathin films. C.X.T., N.S., and I.W. performed the ARPES measurements. K.Y. and T.O. carried out the band calculations. C.X.T. and T.S. finalized the manuscript with inputs from all the authors.

## Competing interests
The authors declare no competing interests.
