## [Peer Review File · Nature Communications]

Reviewers' comments:

Reviewer #1 (Remarks to the Author):

The main point of this paper is the observation of states on the top surface of Pb films grown on the topological insulator TlBiSe₂, and these states resemble the topological surface states of the TlBiSe₂ substrate in vacuum. The authors' interpretation is that these states, with no apparent connection to the electronic structure of Pb films alone, arise from a transfer of the topological surface states from the substrate. They also show that these states exhibit a gap at a temperature below the superconducting transition temperature of Pb. The implication, according to the authors, is that this system is a candidate for topological superconductors.

While the evidence is incomplete, the results are overall quite interesting and suggestive of a potential topological superconductor. A few comments follow:

1. The present study does not unequivocally demonstrate that the observed states are topological. One way to show the topological nature is to map out the dispersion relations (and spin polarization) over the whole Brillouin zone and see how these states connect to the conduction and valence bands. However, the dispersion relations above the Fermi level cannot be mapped by ARPES, which is a basic problem for ARPES. Most ARPES studies would reinforce the case by comparing with theory for the parts that ARPES can actually detect. For the present study, an accurate calculation is unfortunately difficult or infeasible because of the incommensurate nature of the interface. A calculation of the Z₂ parameter and a parity analysis would be good. Without a theoretical comparison, the present case is relatively weak in regard to the topological nature of the states in question. Still, the striking dispersion relations seem to suggest that the authors' assignment might be correct.
2. The authors suggested that their work demonstrates the "topological proximity effect" first discussed by a group involving some of the same coauthors (Ref. 18). It should be noted that the idea of transferring of topological electronic states into adlayers and films had been covered in prior studies. See Wang et al. Phys. Rev. B 87, 035109 (2013) and Phys. Rev. B 87, 235113 (2013). The authors might wish to add a qualitative discussion for the non-experts why such transfer could happen based on adiabatic gap reversal or some other arguments.
3. Prior studies of proximity coupled systems focused on topological insulator films on superconductors. The present work reverses the stacking sequence. The authors argued that this represents a better approach because the topological states in earlier studies are "hard to be accessed by surface-sensitive spectroscopies ..." and "the superconducting helical fermions would be otherwise embedded deep at the interface and are hard to be accessed ..." Actually, some prior studies of topological insulator films on top of superconductors have shown convincing evidence of superconducting gaps in the topological surface states. The gaps may be smaller, but are clearly observed. Of course, it is good to check the complementary system, but the authors statements seem to be a bit of a hard sell. In fact, for a thick film of Pb as in the present study, the topological modes are electrically shorted out by the quantum well states that cross the Fermi level away from the zone center. It is not necessarily a better candidate for applications. On balance, both approaches are interesting and complementary. The discussion can be improved.
4. For the discussion of quantum well states in Pb films, the original study should be cited (instead of or in addition to the later repetition or elaboration): Upton et al., Phys. Rev. Lett. 93, 026802 (2004); Appl. Phys. Lett. 85, 1235 (2004).
5. It should be noted that substrates can have a strong influence on the electronic structure of the

films. The dispersion relations can be modified, and additional states can arise. See, for example, Upton, Phys. Rev. B 71, 033403 (2005) and Liu, Phys. Rev. B 78, 035443 (2008). It is probably useful to emphasize the coherency of the electronic states; both the film and the substrate work together as a composite quantum system to give rise to the effects as observed in the present study. One can imagine that if the Pb film becomes thick in comparison with the electronic coherence length, the substrate effects should vanish. The fact that quantum well states are observed for the film thicknesses studied implies that the coherence length is longer than the film thickness.

6. In the supplementary document, the observed band dispersions were compared with slab calculations. As noted above, the substrates can affect the quantum well states, especially in regard to the interfacial phase shifts. A perfect match in energy is not expected and should not be overly emphasized.

7. The superconducting coherence peak in the ARPES data seems much better defined for the 22 ML film than the other film thicknesses (17 and 12 ML). Why? Is there a significant difference in the damping parameter for thinner films?

8. A worry is that some of the substrate materials might migrate to the top of the Pb film during growth, forming minority compounds on the surface, which could give rise to unusual looking electronic structure. Are Auger or core level data available to show that the film surface contains only Pb, with no traces of Se, Bi, and Tl?

Reviewer #2 (Remarks to the Author):

In this work, the authors have grown superconducting Pb films on topological material TlBiSe₂. Surprisingly, they observed Dirac surface states on the Pb films of ~20 layers (~5 nm) and superconducting energy gaps on the Dirac surface states using high-resolution ARPES. This is different from the previous strategy that topological thin films were grown on superconducting substrates, where superconductivity of the topological surface states was induced by the superconducting proximity effect. This suffers from the limitation of penetration depth of Cooper pairs. This work provides a new way to achieve topological superconductivity. I recommend publishing in Nature Communications. In addition, I have two minor comments.

1. The data quality in Fig. 1 is quite low compared to Figs. 2 and 3. In Fig. 1d I do not see any Dirac bands underneath the dashed guide lines. The Dirac-like band dispersions near the Fermi level in Fig. 1f are also very unclear. The dispersions near the Fermi level in Fig. 1h are blurred. I suggest that the authors replace the data in Fig. 1 with higher quality data.

2. The schematics in Fig. 2g are inconsistent with the experimental results. According to the scenario in Fig. 2g, the experimental observation should be the hybridization of the electronic states of the TlBiSe₂ substrate and the Pb film. There are no any states in the energy range from -0.4 eV to -0.8 eV at Γ in Fig. 2a, whereas band dispersions are present in the energy range in TlBiSe₂ in Fig. 2c. To say more clearly, the results in Fig. 2a,d below -0.4 eV are different from the schematic on the right part in Fig. 2g.

To the comments from Reviewer 1

Reviewer comment: The main point of this paper is the observation of states on the top surface of Pb films grown on the topological insulator TlBiSe₂, and these states resemble the topological surface states of the TlBiSe₂ substrate in vacuum. The authors' interpretation is that these states, with no apparent connection to the electronic structure of Pb films alone, arise from a transfer of the topological surface states from the substrate. They also show that these states exhibit a gap at a temperature below the superconducting transition temperature of Pb. The implication, according to the authors, is that this system is a candidate for topological superconductors. While the evidence is incomplete, the results are overall quite interesting and suggestive of a

potential topological superconductor.

Our response: We thank Reviewer 1 for spending his precious time to read our manuscript and giving us many constructive suggestions to improve our manuscript. We also think it is definitely important to strengthen our main claim by improving our presentation and reinforcing experimental data. Following the thoughtful suggestions from the reviewer, we have revised the manuscript as detailed below. Our point-to-point responses to the reviewer's respective comments are the following.

Reviewer comment: *A few comments follow: 1. The present study does not unequivocally demonstrate that the observed states are topological. One way to show the topological nature is to map out the dispersion relations (and spin polarization) over the whole Brillouin zone and see how these states connect to the conduction and valence bands. However, the dispersion relations above the Fermi level cannot be mapped by ARPES, which is a basic problem for ARPES. Most ARPES studies would reinforce the case by comparing with theory for the parts that ARPES can actually detect. For the present study, an accurate calculation is unfortunately difficult or infeasible because of the incommensurate nature of the interface. A calculation of the Z2 parameter and a parity analysis would be good. Without a theoretical comparison, the present case is relatively weak in regard to the topological nature of the states in question. Still, the striking dispersion relations seem to suggest that the authors' assignment might be correct.*

Our response: We agree with the reviewer on the importance of demonstrating the topological nature of Pb/TlBiSe₂. As pointed out by the reviewer, it is rather difficult to demonstrate the topological nature of this hybrid from the experiment alone because the energy dispersion and its spin texture above the Fermi level cannot be clarified by regular ARPES unless we employ time- and spin-resolved ARPES experiments for the photo-excited states above E_F . This is beyond the capability even for up-to-date ARPES apparatus. Although we tried to calculate the band dispersion of Pb/TlBiSe₂ slab by expanding the in-plane lattice constant of Pb film to hypothetically form a commensurate system, it caused a sizable change in the whole band structure of Pb film, resulting in the band structure totally different from the experiment. To support our main claim by the theory, one needs to find and fabricate a suitable commensurate superconductor/TI hybrid. This is really difficult, but is a big challenge in future. We have elaborated on the situation of slab calculations in the revised manuscript (p. 5, line 1 from the bottom – p. 6, line 4).

Reviewer comment: *2. The authors suggested that their work demonstrates the*

"topological proximity effect" first discussed by a group involving some of the same coauthors (Ref. 18). It should be noted that the idea of transferring of topological electronic states into adlayers and films had been covered in prior studies. See Wang et al. Phys. Rev. B 87, 035109 (2013) and Phys. Rev. B 87, 235113 (2013). The authors might wish to add a qualitative discussion for the non-experts why such transfer could happen based on adiabatic gap reversal or some other arguments.

Our response: We thank the reviewer for letting us know the pioneering theoretical work by Wang et al. These papers were indeed helpful for understanding the topological proximity effect. Our Pb film on TlBiSe₂ is exactly the case depicted in Fig. 1c of the Wang's paper (PRB 87, 235113 (2013)). In this context, the topological proximity effect can be intuitively understood in terms of the adiabatic bulk-band-gap reversal in the real space where the inverted gap in TlBiSe₂ closes throughout the gapless metallic overlayer and starts to open again at the Pb-vacuum interface. We have added a sentence to explain this point in the revised manuscript (p. 5, lines 10-13) by citing literatures pointed out by the reviewer as new refs. 29 and 30.

Reviewer comment: 3. *Prior studies of proximity coupled systems focused on topological insulator films on superconductors. The present work reverses the stacking sequence. The authors argued that this represents a better approach because the topological states in earlier studies are "hard to be accessed by surface-sensitive spectroscopies ..." and "the superconducting helical fermions would be otherwise embedded deep at the interface and are hard to be accessed ..." Actually, some prior studies of topological insulator films on top of superconductors have shown convincing evidence of superconducting gaps in the topological surface states. The gaps may be smaller, but are clearly observed. Of course, it is good to check the complementary system, but the authors statements seem to be a bit of a hard sell. In fact, for a thick film of Pb as in the present study, the topological modes are electrically shorted out by the quantum well states that cross the Fermi level away from the zone center. It is not necessarily a better candidate for applications. On balance, both approaches are interesting and complementary. The discussion can be improved.*

Our response: We share the same opinion with the reviewer that the present approach utilizing the topological proximity effect and the existing approach using the superconducting proximity effect are complementary to each other. The present system may be not necessarily a better system from the application point of view, since the Pb-derived Fermi surface coexists with the Dirac-cone-derived one, so that the topological nature of the superconducting state would be influenced by the Pb-derived Fermi surface.

By taking into account these points, we have improved the discussion part in the revised manuscript (p. 8, lines 13-18).

Reviewer comment: 4. *For the discussion of quantum well states in Pb films, the original study should be cited (instead of or in addition to the later repetition or elaboration): Upton et al., Phys. Rev. Lett. 93, 026802 (2004); Appl. Phys. Lett. 85, 1235 (2004).*

Our response: We thank the reviewer for reminding us of the original works on the quantum well states in Pb films by Upton *et al.* We have cited them in the revised manuscript as new refs. 26 and 27 (p. 4, line 4), and omitted some subsequent repetition/elaboration works from the reference list (ref. 16 in the previous manuscript).

Reviewer comment: 5. *It should be noted that substrates can have a strong influence on the electronic structure of the films. The dispersion relations can be modified, and additional states can arise. See, for example, Upton, Phys. Rev. B 71, 033403 (2005) and Liu, Phys. Rev. B 78, 035443 (2008). It is probably useful to emphasize the coherency of the electronic states; both the film and the substrate work together as a composite quantum system to give rise to the effects as observed in the present study. One can imagine that if the Pb film becomes thick in comparison with the electronic coherence length, the substrate effects should vanish. The fact that quantum well states are observed for the film thicknesses studied implies that the coherence length is longer than the film thickness.*

Our response: We also think it is a good idea to emphasize the coherency of electronic states. Without the coherency, one would be unable to observe the migration of topological Dirac-cone state due to the coupling of electronic states between the Pb film and TlBiSe₂. We expect that the coherence length in Pb film is larger than 22 ML (~6.5 nm) because the topological surface state is observed even in the 22ML film (new Fig. S3f of Supplementary Note 3). We have elaborated on these points by citing the PRB paper by Liu *et al.* as new ref. 31 (p. 6, lines 4-9).

Reviewer comment: 6. *In the supplementary document, the observed band dispersions were compared with slab calculations. As noted above, the substrates can affect the quantum well states, especially in regard to the interfacial phase shifts. A perfect match in energy is not expected and should not be overly emphasized.*

Our response: We thank the reviewer for this expert comment. In fact, we found a small quantitative difference in the location of quantum well states among slab calculations,

experimental data of Pb/TlBiSe₂, and those of Pb/Si, as inferred from a detailed comparison in the energy position of quantum well states between Figs. S1a and S1c. Following the reviewer's suggestion, we have explicitly stated this band mismatch, and discussed its origin in terms of an influence from the substrate (p. 2, lines 4-7 in Supplementary Note 1).

Reviewer comment: 7. The superconducting coherence peak in the ARPES data seems much better defined for the 22 ML film than the other film thicknesses (17 and 12 ML). Why? Is there a significant difference in the damping parameter for thinner films?

Our response: As pointed out by the reviewer, the coherence peak is better seen in the 22 ML film (Fig. 3b) than the 17 ML (Fig. 3c) and 12 ML (Fig. S4a) films. This is also reflected in the broadening factor (damping parameter, Γ) in the numerical fittings with Dynes function at point A on the Pb-derived Fermi surface (0.2, 0.8, and 0.5 meV for the 22-, 17-, and 12-ML films, respectively). While we do not know an exact origin for such a difference, we speculate that the surface quality (such as inhomogeneity and domain size) of Pb film is somehow related to the quasiparticle scattering rate. This conjecture could be further examined by elucidating the relationship between the surface structure and the superconducting gap with low-temperature scanning tunneling microscopy (STM). We have added a few notes on these points in the revised manuscript (last paragraph in Supplementary Note 4).

Reviewer comment: 8. A worry is that some of the substrate materials might migrate to the top of the Pb film during growth, forming minority compounds on the surface, which could give rise to unusual looking electronic structure. Are Auger or core level data available to show that the film surface contains only Pb, with no traces of Se, Bi, and Tl?

Our response: We also think it is important to show the absence of migration of substrate materials to the top of Pb film. We have performed additional ARPES experiments in a wider energy region covering the Tl and

Fig. R1: Comparison of ARPES intensity between pristine TlBiSe₂ and 12ML-Pb/TlBiSe₂ measured in the Tl 5d and Pb 5d core-level region, with the He II α photons ($h\nu = 40.8$ eV).

Pb 5*d* core levels with the He II α photons ($h\nu = 40.8$ eV). We found that the Tl 5*d* core-level peaks located at the binding energy E_B of ~ 12 -16 eV completely disappear after depositing a 12ML Pb film on TlBiSe₂, as shown in Fig. R1. Simultaneously, the Pb 5*d* core-level peaks evolve after Pb deposition. This suggests that the surface measured by ARPES is fully covered with Pb, and the observed Dirac-cone feature in Pb/TlBiSe₂ indeed migrates from the surface of TlBiSe₂. We have elaborated on these points (p. 5, lines 2-5) by adding a new Supplementary Note (Supplementary Note 2 and Fig. S2).

To the comments from Reviewer 2

Reviewer comment: *In this work, the authors have grown superconducting Pb films on topological material TlBiSe₂. Surprisingly, they observed Dirac surface states on the Pb films of ~ 20 layers (~ 5 nm) and superconducting energy gaps on the Dirac surface states using high-resolution ARPES. This is different from the previous strategy that topological thin films were grown on superconducting substrates, where superconductivity of the topological surface states was induced by the superconducting proximity effect. This suffers from the limitation of penetration depth of Cooper pairs. This work provides a new way to achieve topological superconductivity. I recommend publishing in Nature Communications. In addition, I have two minor comments.*

Our response: We thank the reviewer for his thoughtful and constructive comments on our manuscript. We feel very proud to hear that the reviewer appreciates our study and recommend publication of our manuscript. Following his useful suggestions, we have revised the manuscript. In the following, we present our point-to-point responses to the respective comments from the reviewer.

Reviewer comment: *1. The data quality in Fig. 1 is quite low compared to Figs. 2 and 3. In Fig. 1d I do not see any Dirac bands underneath the dashed guide lines. The Dirac-like band dispersions near the Fermi level in Fig. 1f are also very unclear. The dispersions near the Fermi level in Fig. 1h are blurred. I suggest that the authors replace the data in Fig. 1 with higher quality data.*

Our response: Following the useful suggestion from the reviewer, we have updated Fig. 1d, f, h with higher resolution data obtained by additional ARPES experiments on pristine TlBiSe₂, 17 ML Pb film on TlBiSe₂ and on Si(111). The revised ARPES-intensity plots better signify a Dirac-cone-like band in both pristine TlBiSe₂ and 17ML-Pb/TlBiSe₂ as well as its absence in 17ML-Pb/Si(111). It is noted that we had to sacrifice the energy and momentum resolutions to perform simultaneously the Fermi-surface-mapping and the

wide-energy-range ARPES measurements within a limited sample lifetime. This is why the obtained near- E_F band dispersions in Fig. 1d, f, h are not so clear as the high-resolution data in Fig. 2. The present experimental setting where the k cut in Fig. 1 ($\overline{\Gamma M}$; perpendicular to the analyzer slit) is perpendicular to that in Fig. 2 ($\overline{\Gamma K}$; parallel to the analyzer slit) may also cause a finite difference in the near- E_F intensity profile. We have explicitly stated these points in the revised manuscript (caption to Fig. 2).

Reviewer comment: 2. *The schematics in Fig. 2g are inconsistent with the experimental results. According to the scenario in Fig. 2g, the experimental observation should be the hybridization of the electronic states of the TlBiSe₂ substrate and the Pb film. There are no any states in the energy range from -0.4 eV to -0.8 eV at Γ in Fig. 2a, whereas band dispersions are present in the energy range in TlBiSe₂ in Fig. 2c. To say more clearly, the results in Fig. 2a,d below -0.4 eV are different from the schematic on the right part in Fig. 2g.*

Our response: We thank the reviewer for pointing out an inconsistency between the schematics and the experimental band dispersion. As pointed out by the reviewer, the bulk valence-band feature from TlBiSe₂ seen in Fig. 2c is absent in 17ML-Pb/TlBiSe₂ (Fig. 2a). We have rectified the schematic band diagram in the right panel of Fig. 2g and revised the figure caption accordingly, to correctly reflect the experimental results.

REVIEWERS' COMMENTS:

Reviewer #1 (Remarks to the Author):

This reviewer appreciates the authors' strong effort to improve the manuscript. The revised paper is very good.

One lingering issue is the surface chemical composition. In the previous report, it was pointed out that surface migration could happen during film deposition, and it would be good to check to make sure that the surface of the Pb film does not contain Tl, Bi, and Se. The phenomenon of surface segregation or migration is known from many prior studies. The biggest worry is Se segregation, which might give rise to electronic structure resembling a free electron band near the zone center. To the authors' credit, they now include a figure (S2) in the supplementary document to show that the Tl signal is completely covered up by the film, but this does not say anything about Se. Ideally, the authors should apply Auger or XPS, but this was apparently not available in their system.

It would seem too much to ask for the authors to do further measurements. This reviewer is willing to accept the explanation, but the authors can perhaps revise their statement in the supplementary document to clarify that the measurements shown in Fig. S2 might suggest a clean interface/surface (or consistent with a clean surface), but it is not a complete proof as the data range does not cover signals from Se.

Another minor comment to the authors for their consideration, not relevant to acceptance, is that one could potentially use a larger in-plane unit cell to achieve an approximate lattice match for the calculations. Doing so avoids the need to do a drastic compression or expansion of the Pb lattice. This is a standard trick that has been employed in numerous prior studies.

The paper should be acceptable with the minor revision suggested above. Congratulations to the authors for a nice experimental study.

Reviewer #2 (Remarks to the Author):

I am satisfied with the revision of the manuscript, especially for my comments.

I have one suggestion about the discussion of the topological nature of the Dirac bands. Another reviewer is also concerned about this. The lower branch of the Dirac bands is connected to the uppermost M-shape band, which is below the Fermi level along Γ -M. While the dispersion of the Dirac bands above the Fermi level is unknown based on the current data, one can deduce that the upper branch of the Dirac bands is connected to the conduction band, since it only passes through the Fermi level one time between Γ and M.

To the comments from Reviewer 1

Reviewer comment: *This reviewer appreciates the authors' strong effort to improve the manuscript. The revised paper is very good.*

Our response: We thank Reviewer 1 for spending his precious time again to read our manuscript. We are pleased to know that the reviewer satisfied our revision. Following the suggestions from the reviewer, we have revised the manuscript as detailed below.

Reviewer comment: *One lingering issue is the surface chemical composition. In the previous report, it was pointed out that surface migration could happen during film deposition, and it would be good to check to make sure that the surface of the Pb film does not contain Tl, Bi, and Se. The phenomenon of surface segregation or migration is known from many prior studies. The biggest worry is Se segregation, which might give rise to electronic structure resembling a free electron band near the zone center. To the authors' credit, they now include a figure (S2) in the supplementary document to show that the Tl signal is completely covered up by the film, but this does not say anything about Se. Ideally, the authors should apply Auger or XPS, but this was apparently not available in their system.*

It would seem too much to ask for the authors to do further measurements. This reviewer is willing to accept the explanation, but the authors can perhaps revise their statement in the supplementary document to clarify that the measurements shown in Fig. S2 might suggest a clean interface/surface (or consistent with a clean surface), but it is not a complete proof as the data range does not cover signals from Se.

Our response: We thank the reviewer for accepting our explanation. Following the suggestion from the reviewer, we have stated that the core-level data might suggest a clean interface/surface, and also explicitly stated that the data range does not cover the Se core levels (p. 3, lines 7-10 in Supplementary Note 2).

Another minor comment to the authors for their consideration, not relevant to acceptance, is that one could potentially use a larger in-plane unit cell to achieve an approximate lattice match for the calculations. Doing so avoids the need to do a drastic compression or expansion of the Pb lattice. This is a standard trick that has been employed in numerous prior studies.

The paper should be acceptable with the minor revision suggested above. Congratulations to the authors for a nice experimental study.

Our response: Following the reviewer's suggestion, we have stated that a calculation that uses a larger in-plane unit cell might be useful to achieve an approximate lattice match between Pb and TlBiSe₂ (p. 6, lines 8-9).

To the comments from Reviewer 2

Reviewer comment: *I am satisfied with the revision of the manuscript, especially for my comments.*

I have one suggestion about the discussion of the topological nature of the Dirac bands. Another reviewer is also concerned about this. The lower branch of the Dirac bands is connected to the uppermost M-shape band, which is below the Fermi level along Γ -M. While the dispersion of the Dirac bands above the Fermi level is unknown based on the current data, one can deduce that the upper branch of the Dirac bands is connected to the conduction band, since it only passes through the Fermi level one time between Γ and M.

Our response: We are pleased to hear that Reviewer 2 satisfied our revision. Following the reviewer's suggestion, we have explicitly stated that the upper branch of the Dirac-cone-like band would be connected to the quantized conduction band above E_F because it only crosses the Fermi level once between Γ and M (p. 5, lines 14-16).

Finally, we thank again both the reviewers for their thoughtful and constructive suggestions to improve our manuscript.